Physiochemical screening of road avenue plants in better landscape management of highly polluted urbanized city (Lahore), Pakistan

Munam Bushra
Muhammad Sohaib dr.sohaibmuhammad@gcu.edu.pk
Tayyab Muhammad
Hanif Hafiza Komal
Majeed Mahrukh
Nawaz Hassan
Khan Muhammad Jawad Tariq
Faisal Summiya
Hasnain Muhammad
Malik Sarah Maryam
Bilal Muhammad
Zahid Muhammad
Department of Botany, Government College University Lahore , Lahore , Punjab , Pakistan
Wang Xinfeng
Electronic publication date: 2025 Oct 31
Publication date: 2025
Volume: 13
Electronic Location ID: e20121
Received 2024 Aug 27; Accepted 2025 Sep 1
Copyright: ©2025 Munam et al.
Copyright year: 2025
Copyright holder: Munam et al.
License: This is an open access article distributed under the terms of the Creative Commons Attribution License, which permits unrestricted use, distribution, reproduction and adaptation in any medium and for any purpose provided that it is properly attributed. For attribution, the original author(s), title, publication source (PeerJ) and either DOI or URL of the article must be cited.
License URL: https://creativecommons.org/licenses/by/4.0/

Keywords: Green belt, Pollution, Urbanized city, Road avenue plants, Physiochemical attributes, Photosynthetic rate, Stomatal conductance, Transpiration rate, Chlorophyll content, Pollution mitigation

Funding: The authors received no funding for this work.

==============================
Lahore has been consistently ranked as the world’s most polluted city. Because of combative ideas to construct highways, underpasses and flyovers, Lahore had lost a remarkable percentage of its tree cover over the past 15 years. The present study focuses on the outcomes of rapidly increasing air pollution on roadside vegetation. In current study, species such as Alstonia scholaris L., Bougainvillea spectabilis Willd., Dalbergia sissoo Roxb. Eucalyptus globulus Labill., Ficus virens Aiton, Ficus benjamina L., Ficus religiosa Linn., Morus alba L., Murraya paniculata L., Putranjiva roxburghii Wall., Polyalthia longifolia Sonn., Rubia tinctorum L. found on the seven busiest roads of Lahore were selected (on the basis of traffic densities) for biomonitoring. These plants were selected due to their prevalence and commonly occurrence on these selected roads. Variation on biochemical parameters like chlorophyll a, b, total chlorophyll content & carotenoids and physiological parameters like stomatal conductance, transpiration rate and photosynthetic rate were found in triplicate. By analyzing these parameters quality of air and health of plants can also be assessed. In this study the dust load was maximum on the leaves of Alstonia scholaris L. (0.02 ± 0.005), Ficus religiosa Linn. (0.02 ± 0.003), and Morus alba L. (0.02 ± 0.003) Reduction in chlorophyll was noticed in Alstonia scholaris L. (0.44 ± 0.22) and Polyalthia longiflia Sonn. (0.41 ± 0.22) while the chlorophyll concentration of Eucalyptus globulus Labill. (0.71 ± 0.16), followed by Ficus benjamina L. (0.80 ± 0.25), Ficus religiosa Linn. (0.81 ± 0.30), Ficus virens Aiton. (0.64 ± 0.22), Morus alba L. (1.80 ± 0.27) and Putranjiva roxburghii Wall. (2.55 ± 0.43), was higher at polluted sites. The reduction in carotenoid content was found in Murraya paniculata L. (4.12 ± 2.18) while it was highest in Eucalyptus globulus Labill. (9.12 ± 0.71) Due to the pollution stress the changes in photosynthetic rate of Alstonia scholaris L. (25.36 ± 13.10), Bougainvillea spectabilis Willd. (34.37 ± 19.92), Dalbergia sissoo Roxb. (28.23 ± 11.25), Murraya paniculata L. (26.80 ± 7.75), Polyalthia longifolia Sonn. (42.27 ± 22.87), and Rubia tinctorum L. (30.60 ± 4.07) was observed. The current research distinctly signifies Eucalyptus globulus Labill., Ficus benjamina L., Ficus religiosa Linn., Ficus virens Aiton., Morus alba L. and Putranjiva roxburghii Wall., have capability to hold on the stress triggered by roadside pollutants. The findings are useful to urban green space landscapers in harsh climates as they choose appropriate species that can offer a variety of ecosystem services, such as resistance to air pollution and lowering of temperature without compromising plant survival.

Introduction

Air pollution is one of the most serious environmental issue of the world now a days, which need immediate attention to be addressed at a global level by adopting friendly strategies. The majority of urban locations in developing as well as developed countries have air pollution concerns worldwide (Singh et al., 2020). The rapidly rising population, random urbanization, industrialization and extensive automobile use are contributing to a range of environmental challenges. Urban spaces are marked by unplanned developmental activities and lack of infrastructural strategies, contributed in deterioration of the quality of the environment with serious consequences for human well-being and regional biodiversity, particularly plants in urban environments (Skrynetska et al., 2018). The sharp rise in number of vehicles, industrial exhaust and the reduction in plant cover in metropolitan areas, are the primary contributors to air quality issues in cities (Sass et al., 2017). Air pollution caused by vehicles is an extremely sensitive threat to the environment for human society (Mukherjee et al., 2019). The principal contributions to air pollution concentrations in the atmosphere include anthropogenic and certain natural activities (Vardhan, Kumar & Panda, 2019). Among the largest and most common sources of these heavy metals are vehicular emissions that can have a negative impact on roadside vegetation (Sarhan, Elhafeez & Bashanday, 2021).

The growing number of vehicles and particulate matter loads contributes significantly to both regional and global environmental pollution. Industries emit massive amounts of harmful contaminants and toxins into the natural environment, including carbon monoxide (CO), particulate matter (PM), and hydrocarbons, causing greenhouse emissions (Munsif et al., 2021). As one of the air contaminants, particulate matter in the atmosphere (PM2.5) is projected to cause 3.3 million premature deaths per year primarily in Asia and has a variety of harmful consequences on the well-being of humans (Nowak et al., 2018). Vehicular exhaust causes environmental harm in countries like Pakistan due to inadequately maintained automobiles and the use of low-quality fuel. Carbon dioxide (CO2), nitrous oxide (NOx), CO, sulfur dioxide (SO2), hydrocarbons (HCs), PM and volatile organic compounds (VOCs) are all emitted by vehicles and account for 60% to 70% of metropolitan air pollution (Uka, Belford & Hogarh, 2019). These toxic substances directly contributed in bad health of the ecosystem, mainly the plants and soil (Nazarpour et al., 2019; Kaur et al., 2022).

Hence, biomonitoring of air pollution is crucial for urban restoration of ecosystems as pollutants adversely affect the environment (Sekhar & Sekhar, 2019). Various researchers have recognized the significance of plants in air pollution reduction (He et al., 2020). In addition to this, roadside plants offer an essential role in reducing air pollution since their enzymatic activity, physiological and anatomical characteristics aid in establishing and maintaining mitigation potential towards vehicular emissions (Khalil et al., 2022; Mahrukh Awan et al., 2023). Modification of leaf anatomical features play a pivotal role to control plant physiological activity (Kumar et al., 2022), including photosynthetic rate, transpiration rate, and chlorophyll concentration (Khalid et al., 2019). While these physicochemical changes are evidently marked on leaves of the plants under stress of pollution than any other part of the plant i.e., roots and stem (Wei et al., 2021). Plants thrive in polluted environments can provide an insight into the physiological, metabolic, and genetic mechanisms as tolerant or sensitive plant under stressed environmental conditions. Subsequently can be helpful in devising new strategies for preserving plant species against pollution (Roy, Battacharya & Kumari, 2020; Sahu, Basti & Sahu, 2020). Various organizations of life like fungi, lichens, tree rings, leaves, and barks of trees are studied by various researchers of the world. Vulnerable species, such as lichens, cannot thrive in densely crowded commercial and metropolitan environments. As a result, trees or shrubs may be used as ecological indicator species in urban settings to assess air quality (Ghafari et al., 2021) and to develop the green belts of specific plant species around urban areas to reduce the impact of airborne pollutants (Bharti, Trivedi & Kumar, 2018).

Inevitably, this region of the world has received the least attention and lacks an adequate air quality monitoring network. It is critical to analyze the degraded ambient air quality on a regular basis and to assess the effectiveness of mitigation strategies (Ghorbanian et al., 2020). The Air Quality Index (AQI) does not fulfill World Health Organization (WHO) air quality recommendations in many cities of Pakistan such as Lahore, Peshawar, Karachi and Islamabad, especially throughout the winter and autumn months (PAQI, 2018). As the capital of the Punjab province, Lahore has the highest aggregation of transportation in the city, contributing to the growth in air pollution. In the last decade, Lahore has been regarded as one of the most polluted cities in the world with variable unhealthy AQI, this year AQI recorded as 394. This air quality is measured with various concentrations of the pollutants like PM2.5 (Particulate Matter2.5), PM10 (coarse particulate matter, NO2 (nitrogen dioxide) and O3 ozone and AQI above 150 is considered very unhealthy (Staff Reporter, 2024). It was evidently noticed that there were several sectors which had the main contribution in drastic air quality of the city, these sectors include domestic, commercial, waste burning, agriculture (crop residue burning), industrial and transport with 0.11, 0.14, 3.6, 3.9, 4.07 and 83.15% share respectively. Among these sectors, the transport sector has the highest contributions in terms of air pollutants viz., CO (73%), non metallic volatile organic carbon (14%), sulfur oxide (SOx) (7%), NOx (4%), total suspended particles (1%) and PM2.5 (1%). Overall, the city of gardens (Lahore) is now facing the hazardous AQI index continuously from last few years (Ilyas & Nissar, 2022).

The purpose of present research work was to determine (1) the existing plant diversity of the city along busiest roads, (2) selection of the dominant plant species, (3) determination of the automobile exhaust (amount of dust), (4) evaluation of the response of plants through their growth (leaf width, length & area) and physiochemical attributes

Materials & Methods

The influences of vehicular pollution upon growth and physicochemical features of plants were investigated by selecting multiple roads of Lahore city.

Study site

Seven busiest roads of the city were chosen based on traffic density, this data was obtained from Traffic Engineering and Planning Agency (TEPA), Lahore Pakistan. The given data in Table 1 for each road was calculated into average of the different types of the vehicles for two peak hour periods, each consisting of a 4 h period (7:00–11:00 am & 4:30–8:30 pm) of the day. Selected roads for sampling (Study Sites S) of the plants were Scheme Morr to Dubai Chowk (Allama Iqbal Town Road, S1), Dubai Chowk to Bhekhewal Morr (Allama Iqbal town Road, S2), Bhekhewal Morr to Karim Market Round about (Fazal-e- Haq, S3), Karim Market round about to Karim Market Signal (Moullana Hasrat Mohani Road, S4), Karim Market Signal to Bhekhewal Morr (WahdatRoad-1, S5), Karim Market Signal to Elephant Link Road (Whadat Road-2, S6) and Karim Market Signal to Jinnah Hospital (Site-7, S7), Control site plant samples were collected from less contaminated locations, such as neighboring parks. The samples were collected in triplicates from the selected roadside plants.

Floristic composition

A survey of seven selected roadways was done to determine the plant species on study sites. The available literature was then used to identify the plant species (Nasir & Ali, 1970-89; Ali & Nasir, 1990; Ali & Qaiser, 1992). Subsequently, plants were selected depending on the percentage and uniformity throughout the route and sampling was carried out from February to June, 2022, when the plants were at their peak of growth (Zheng, Ge & Hao, 2013; Liu & Zhang, 2020).

Selection of plant species

Plant species were selected on the basis of dominance of the species (i.e., the no. of plant species) and their common occurrence at each road side (Muller-Dombois & Ellenberg, 1974). The dominantly occurring plant species were: Alstonia scholaris L., Bougainvillea spectabilis Willd., Dalbergia sissoo Roxb. Eucalyptus globulus Labill., Ficus benjamina L., Ficus virens Aiton., Ficus religiosa Linn., Morus alba L., Murraya paniculata L., Putranjiva roxburghii Wall., Polyalthia longifolia Sonn. and Rubia tinctorum L. as shown in Table 2 with their habit and taxonomic classification.

Table 1 Variable traffic density on some selected road sites of Lahore city (Punjab), Pakistan.

Time period (s)	Selected sites with density of vehicles*	
	*S1	*S2	*S3	*S4	*S5	*S6	*S7	
7:00–11:00 am	13,483	18,266	12,342	14,556	20,823	11,209	13,575	
4:30–8:30 pm	11,488	16,703	13,517	13,998	15,546	11,785	14,708	
Notes.

Vehicles* include Motorcycles & Scooters, Rickshaw, Qingqi Rickshaw, Car, Wagon, Mini & Large Buses and Trucks.

*S1 to *S7 includes Scheme Morr to Dubai Chowk (Allama Iqbal Town Road, S1), Dubai Chowk to Bhekhewal Morr (Allama Iqbal town Road, S2), Bhekhewal Morr to Karim Market Round about (Fazal-e- Haq, S3), Karim Market round about to Karim Market Signal (Moullana Hasrat Mohani Road, S4), Karim Market Signal to Bhekhewal Morr (WahdatRoad-1, S5), Karim Market Signal to Elephant Link Road (Whadat Road-2, S6) and Karim Market Signal to Jinnah Hospital (Site-7, S7).

Table 2 Taxonomic illustration of selected plants in urban ecosystem of Lahore (Pakistan).

Sr.no	Scientific name	Family	Common name	Habit	
1.	Alstonia scholaris L.	Apocynaceae	Blackboard or devil tree	Perennial, Evergreen Large Trees	
2.	Bougainvillea spectabilis Willd.	Nyctaginaceae	Paperflower	Woody climber, perennial shrub	
3.	Dalbergia sissoo Roxb.	Fabaceae	North Indian rosewood or shisham	Deciduous tree	
4.	Eucalyptus globulus Labill.	Myrtaceae	Tasmanian bluegum	Annual or seomtimes perennial Evergreen tree	
5.	Ficus virens Aiton.	Moraceae	White fig	Medium sized tree, perennial evergreen tree	
6.	Ficus benjamina L.	Moraceae	Weeping fig	Perennial evergreen shrub or tree	
7.	Ficus religiosa L.	Moraceae	Peepal	Perennial and deciduous tree	
8.	Morus alba L.	Moraceae	White mulberry	Deciduous tree or shrub	
9.	Murraya paniculata L.	Rutaceae	Orange Jasmine, Mock Lime, China Box	Perennial shrub or small tree	
10.	Putranjiva roxburghii Wall.	Putranjivaceae	Childlife tree, Lucky Bean Tree 	Moderate sized, evergreen tree 	
11.	Polyalthia longifolia Sonn.	Annonaceae	False Ashoka	Evergreen tree	
12.	Rubia tinctorum L.	Rubiaceae	Rose madder or common madder 	Evergreen herbaceous Perennial	

Design of experiment

The sampling from the selected seven sites were carried out from the dominant 12 plant species of these road sites by following the completely randomized design. Three trees of each selected plant species were targeted at a variable distance (about 15–20 m) and the leaf samples were collected in triplicate (nine leaf samples of each species). Before collection of the leaf samples, the physiological parameters were assessed through Infrared Gas Analyzer (IRGA) and for growth attributes of plant, dust content and biochemical attributes the leaf samples were packed in polythene bags and transported to the laboratory for further above mentioned analysis.

Dust content (g)

By using weighing balance, filter paper was weighed. Dust from leaves was removed with help of camel hairbrush and reweighed filter paper. Amount of dust was calculated by subtracting both values (filter paper without dust and filter paper with dust) (Hussain, Ilahi & Rashid, 1989).

Growth attributes of plants

Growth attributes of the selected plants under automobile exhaust pollution were assessed with respect to leaf length (mm), leaf width (mm) and leaf area index (mm2) by using leaf area meter (Hussain, Ilahi & Rashid, 1989).

Physiological assessments

The Infrared Gas Analyzer (IRGA, LCA-4) was used to assess physiological features such as photosynthetic rate (µmol m−2s−1), transpiration rate (mol m−2s−1) and stomatal conductance (mol m−2s−1) (Muhammad et al., 2014).

Biochemical assessments

In order to determine carotenoids content, chlorophyll a, b and total chlorophyll, the biochemical parameters assessed were as follows:

Total chlorophyll content (mg/g)

Chlorophyll content was determined by using the method of acetone extraction (Singh et al., 1991). Two grams of fresh leaves were taken and crushed in 20 ml of 80% acetone by using pestle and mortar. Through filtration, extract of plant leaves was obtained. The absorbance of the filtrate was determined with the help of spectrophotometer at the wavelengths of 663 nm and 645 nm. The whole process was repeated for each plant sample.

To determine the total chlorophyll content following equation was used: (1) Chl. a=A663×0.0127+A645×0.00269×10×W

(2) Chl. b=A645×0.0229+A663×0.00468×10×W

(3) Total chlorophyll content=20.2×A645+8.02×A663×V/1000×W

where, A663 = Absorbance at 663 nm, A645 = Absorbance at 645

V= Total volume of extract, W= Fresh weight of leaves (g).

Carotenoid content (mg/g)

One g leaves were grinded by using 100% acetone. Through filtration extract of plants were obtained and raise the final volume of extraction solution up to 50 ml. The carotenoid content was determined by spectrophotometer at the wavelength of 440.5 nm (Zofia, Kmiecik & Korus, 2006).

To measure the carotenoid content following equation was used: (4) Carotenoids contents:V×383×As−Ab/100×W

where, ‘V’ = volume

‘383’ = carotenoids extinction coefficient

‘As’ = sample absorbance

‘Ab’ = cuvette error

‘W’ = weight of sample (g)

Statistical analysis

The samples of the plants collected from each road side were tested against growth and physiochemical attributes and their statistical significance of the findings were calculated by comparing the means of the attributes through Analysis of the Variance (ANOVA) after Steel, Torrle & Dicky (1997) and their possible response against the particulate dust accumulation through principal component analysis (PCA) after (Mackiewicz & Ratajezak, 1993). While the association among various variables used in this study was developed through multiple correlation technique i.e., heat map by following (Tiranakwit et al., 2023).

Results

Total numbers of 85 plants species were recorded in floristic composition by roadside surveys. For this research work, 12 dominant and common plants species found on all the selected road sites were chosen. On the seven selected study sites, the common plants of urban landscape were: A. scholaris L., B. spectabilis Willd., D. sissoo Roxb., E. globulus Labill, F. virens Aiton., F. benjamina L., F. religiosa L., M. alba L., M. paniculata L., P. roxburghii Wall., P. longifolia Sonn., R. tinctorum L. These plants were under culminating pressure of bad ambient air quality of the Lahore city.

A. scholaris, F. religiosa, and M. alba (0.02 ± 0.005, 0.02 ± 0.003, and 0.02 ± 0.003) showed the highest dust deposition (g), respectively because of their increased canopy cover, which enhances dust trapping capability. This observation is consistent with large leaf area indexes (LAI) mm2, particularly in A. scholaris (1,693 ± 63, control site) and P. longifolia (1,071.2 ± 139.6, polluted site). M. alba had the highest photosynthetic rate (63.01 ± 9.23 at polluted site) µmol m−2s−1, indicating that increased leaf area promotes photosynthetic activity. This shows a positive relationship between canopy cover, leaf area, and photosynthetic efficiency in polluted environments. In comparison, species with minimal canopy coverage, such as F. benjamina and M. paniculata (0.011 ± 0.0008 and 0.011 ± 0.0003) showed the lowest dust deposition respectively, in the polluted site. These species, particularly M. paniculata (88.86 ± 19.7, polluted site), have reduced LAI. This shows a direct link between LAI and the plant’s ability to trap dust (Table 3). F. religiosa had the maximum stomatal conductance (0.62 ± 0.08 versus 0.58 ± 0.08 in polluted and control sites) mol m−2s−1, indicating its ability to maintain gas exchange even under adverse conditions. This character, combined with its high dust deposition, indicates that F. religiosa is well-suited to polluted habitats. D. sissoo (0.03 ± 0.003, polluted site) and A. scholaris (0.07 ± 0.03, control site) had the lowest stomatal conductance, indicating decreased gas exchange and lower stress tolerance. Transpiration rates (mol m−2s−1) showed F. benjamina had the highest values (0.86 ± 0.30 in polluted and 0.85 ± 0.51 in control sites), which may indicate its increased upward water movement through xylem. However, P. longifolia had the lowest transpiration rate (0.10 ± 0.04, polluted site), suggesting limited water loss, an adaptation to conserve water under stress. A. scholaris and E. globulus had the lowest photosynthetic rates (25.36 ± 13.10 and 27.64 ± 14.52, respectively), indicating low tolerance to pollution despite high LAI (Table 4).

Table 3 Amount of dust (g) captured and leaf area index (mm2) of the selected tree species at 0.05 significance level (p = 0.05).

Sr. no.	Plant species	Amount of dust (g)	Leaf area index (mm 2 )	
		Pol. s	Cont. s	Pol. S	Cont. s	
1.	A. scholaris	0.02b ± 0.005	0.004a ± 0.0005	543.13d ± 306.6	1,693d ± 63	
2.	B. spectabilis	0.015ab ± 0.003	0.004a ± 0.0003	312.9a ± 11.59	532.33ab ± 4.70	
3.	D. sissoo	0.014ab ± 0.002	0.003a ± 0.0006	271.04a ± 66.50	414.7ac ± 43.006	
4.	E. globulus	0.014ab ± 0.004	0.007a ± 0.001	294.73a ± 8213	436.6ac ± 22.48	
5.	F. benjamina	0.011a ± 0.0008	0.003a ± 0.0003	257.84ab ± 12.10	341.8c ± 3.190	
6.	F. religiosa	0.02b ± 0.003	0.004a ± 0.0005	201.93ab ± 450.4	312.53c ± 31.73	
7.	F. virens	0.016ab ± 0.003	0.004a ± 0.000	958.82c ± 380.92	1,670d ± 23.00	
8.	M. alba	0.02b ± 0.003	0.006a ± 0.0005	997.8c ± 25.175	1,172.86e ± 58.69	
9.	M. paniculata	0.011a ± 0.0003	0.004a ± 0.0003	88.86b ± 19.7	97.96f ± 4.88	
10.	P. roxburghii	0.012ab ± 0.0003	0.004a ± 0.0003	496.66d ± 50.33	611.66b ± 35.86	
11.	P. longifolia	0.016ab ± 0.004	0.005a ± 0.000	1,071.2c ± 139.6	1,182.5e ± 113.5	
12.	R. tinctorum	0.013ab ± 0.006	0.007a ± 0.0013	306.55a ± 16.93	351.11c ± 13.55	
Notes.

Polluted site, (Pol. s).

Control site (Cont. s).

Different lowercase letters showed significant variation among mean values of aforementioned parameters at 0.05 significance level.

Table 4 Physiological evaluation of plant species (polluted and control sites) at 0.05 significance level.

Sr. no.	Plant species	Stomatal conductance
(mol m−2s−1)	Transpiration rate
(mol m−2s−1)	Photosynthetic rate
(μ mol m−2s−1)	
		*Pol. s	**Cont. s	Pol. s	Cont. s	Pol. s	Cont. s	
1.	A. scholaris	0.04a ± 0.02	0.07a ± 0.03	0.17ab ± 0.09	0.25ad ± 0.14	25.36a ± 13.10	34.07ab ± 17.68	
2.	B. spectabilis	0.05a ± 0.01	0.15a ± 0.006	0.18ab ± 0.04	0.48abc ± 0.13	34.37ab ± 19.92	35.88ab ± 8.12	
3.	D. sissoo	0.03a ± 0.003	0.08a ± 0.03	0.21abc ± 0.04	0.32abd ± 0.01	28.23a ± 11.25	51.58ab ± 20.53	
4.	E. globulus	0.09abc ± 0.02	0.08a ± 0.05	0.26abc ± 0.08	0.22ad ± 0.11	27.88a ± 9.83	27.64a ± 14.52	
5.	F. benjamina	0.17bd ± 0.04	0.16a ± 0.09	0.86d ± 0.30	0.85e ± 0.51	56.60bc ± 13.34	53.73ab ± 26.89	
6.	F. religiosa	0.62e ± 0.08	0.58b ± 0.08	0.67de ± 0.11	0.59bce ± 0.25	60.22bc ± 15.72	50ab ± 25	
7.	F. virens	0.19d ± 0.03	0.17a ± 0.08	0.42cfg ± 0.11	0.41abc ± 0.21	44.41abc ± 12.22	44.26ab ± 21.83	
8.	M. alba	0.15bcd ± 0.01	0.14a ± 0.008	0.54efg ± 0.01	0.48abc ± 0.012	63.01c ± 9.23	59.73b ± 1.01	
9.	M. paniculata	0.04a ± 0.005	0.13a ± 0.01	0.24abc ± 0.09	0.62ce ± 0.04	26.80a ± 7.75	45.14ab ± 2.89	
10.	P. roxburghii	0.17bd ± 0.005	0.15a ± 0.01	0.34bcf ± 0.03	0.32abd ± 0.01	56.90bc ± 12.19	57.91b ± 6.13	
11.	P. longifolia	0.08ac ± 0.01	0.09a ± 0.04	0.10a ± 0.04	0.13d ± 0.06	42.27abc ± 22.87	43.89ab ± 24.00	
12	R. tinctorum	0.08ac ± 0.05	0.17a ± 0.09	0.59eg ± 0.24	1.61f ± 0.22	30.60ab ± 4.07	40.12ab ± 0.22	
Notes.

Polluted site, (Pol. s).

Control site (Cont. s).

Different lowercase letters showed significant variation among mean values of aforementioned parameters at 0.05 significance level.

Control sites showed higher Chl. a, Chl. b, Total Chl. and Carotenoids contents compared to polluted sites for most species except E. globules and M. alba which indicates pollution has affected photosynthetic efficacy of studied plants. In polluted site, Chl. a content was found to be higher in M. alba (0.76 ± 0.09) compared to the other selected plant species. While in control site, M. paniculata (0.73 ± 0.03) had highest Chl. a concentration among other selected plants. E. globulus (0.24 ± 0.09 at polluted site, 0.23 ± 0.12 at control site) showed lowest Chl. a concentration among other plant species. P. roxburghii (for Chl. b = 1.20 ± 0.12 at polluted site, 1.35 ± 0.10 at control site, for Total Chl. = 2.55 ± 0.43 at polluted site, 2.33 ± 0.07 at control site) had higher levels of Chl. b and Total Chl. (mg/g) in polluted as well as in control site as compared to other plant species. This shows that this species has better photosynthetic capacity, resulted the faster growth and output. High total chlorophyll values indicate its resilience to environmental conditions. Lowest values of Chl. b in polluted and control site both were observed in E. globules (0.32 ± 0.10 at polluted site, 0.30 ± 0.15 at control site). Carotenoids content (mg/g) was found to be higher in E. globulus (9.12 ± 0.71 for polluted site) and lower in M. paniculata (4.12 ± 2.18 at polluted site, 4.45 ± 1.84 at control site) compared to other plant species. Higher carotenoid level in this species may indicate better protection against oxidative stress, which is critical in environments with intense light or other environmental stressors. The diversity in biochemical composition observed in the radar chart reflects species-specific adaptations to environmental conditions (Fig. 1). While at control site, D. sisso (11.18 ± 3.31) had the highest value of carotenoids content (Table 5). These relationships indicate that species with higher canopy coverage, effective stomatal conductance, and transpiration rates, such as F. religiosa and M. alba, are more adaptable and function better in polluted environment. In contrast, species with lower LAI, such as M. paniculata and D. sissoo, are less able to survive pollution stress.

Figure 1 Radar map visualization for comparing carotenoid and chlorophyll level in different plant species.

A. sch, Alstonia scholaris; B. spe, Bougainvillea spectabilis; D. sis, Dalbergia sissoo; E. glo, Eucalyptus globulus; F. ben, Ficus benjamina; F. rel, Ficus religiosa; F. vir, Ficus virens; M. alb, Morus alba; M. pan, Murraya paniculata; P. rox, Putranjiva roxburghii; P. lon, Polyalthia longifolia; R. tin, Rubia tinctorum.

Table 5 Biochemical assessment of species from polluted and control sites at 0.05 significance level.

Sr. No.	Plant species	Chl. a	Chl. b	Total chlorophyll
(mg/g)	Carotenoids
(mg/g)	
		Pol. s	Cont. s	Pol. s	Cont. s	Pol. s	Cont. s	Pol. s	Cont. s	
1.	A. scholaris	0.30ab ± 0.15	0.38aef ± 0.19	0.34de ± 0.17	0.55bef ± 0.27	0.44cd ± 0.22	1.20cd ± 0.62	5.94abf ± 0.07	6.88g ± 2.97	
2.	B. spectabilis	0.43ab ± 0.14	0.52abcde ± 0.16	0.72ab ± 0.28	1.14a ± 0.31	1.29a ± 0.59	2.21a ± 0.73	6.04ab ± 0.88	9.07a ± 2.23	
3.	D. sissoo	0.56ac ± 0.08	0.64abcd ± 0.01	0.70abc ± 0.10	0.75bcd ± 0.10	0.56bc ± 0.13	0.82bcd ± 0.01	7.15cd ± 3.03	11.18b ± 3.31	
4.	E. globulus	0.24b ± 0.09	0.23f ± 0.12	0.32d ± 0.10	0.30e ± 0.15	0.71bcd ± 0.16	0.65bcd ± 0.29	9.12e ± 0.71	6.20c ± 2.36	
5.	F. benjamina	0.36ab ± 0.14	0.44abdef ± 0.22	0.58abcde ± 0.24	0.64bcd ± 0.32	0.80b ± 0.25	0.75b ± 0.38	5.25af ± 1.64	5.22d ± 2.77	
6.	F. religiosa	0.32ab ± 0.16	0.40aef ± 0.20	0.40cde ± 0.21	0.55bef ± 0.27	0.81b ± 0.30	0.65b ± 0.42	5.00fg ± 2.01	5.82e ± 2.91	
7.	F. virens	0.26b ± 0.14	0.30ef ± 0.19	0.44acde ± 0.22	0.38ef ± 0.16	0.64bcd ± 0.22	0.61bcd ± 0.31	7.76c ± 2.91	7.17fg ± 3.88	
8.	M. alba	0.76c ± 0.09	0.67bcd ± 0.03	0.88b ± 0.13	0.83cd ± 0.04	1.80e ± 0.27	1.36e ± 0.07	7.91c ± 1.61	7.18fg ± 1.07	
9.	M. paniculata	0.51abc ± 0.01	0.73c ± 0.03	0.65abce ± 0.11	0.87c ± 0.07	0.75bc ± 0.12	1.14bc ± 0.11	4.12g ± 2.18	4.45h ± 1.84	
10.	P. roxburghii	0.56ac ± 0.10	0.71bc ± 0.06	1.20f ± 0.12	1.35a ± 0.10	2.55f ± 0.43	2.33f ± 0.07	7.60c ± 2.02	7.48f ± 0.83	
11.	P. longifolia	0.33ab ± 0.18	0.41adef ± 0.21	0.55acde ± 0.27	0.59bdf ± 0.27	0.41d ± 0.22	0.44d ± 0.25	6.29bd ± 1.86	7.43f ± 3.72	
12	R. tinctorum	0.41ab ± 0.03	0.51abcde ± 0.008	0.53acde ± 0.09	0.63bcdf ± 0.06	0.67bcd ± 0.15	0.98bcd ± 0.16	7.84c ± 0.39	9.69i ± 0.21	
Notes.

Polluted site, (Pol. s).

Control site (Cont. s).

Different lowercase letters showed significant variation among mean values of aforementioned parameters at 0.05 significance level.

One-way ANOVA was performed at 0.05 significance level (p = 0.05) to analyze variations in the means of physicochemical parameters from plant species at both polluted and control sites. Significant variations among the means of the variables were noted as shown with their statistics in Table 6.

Table 6 One-way ANOVA for various physicochemical parameters between species at control and polluted sites.

Parameter	Site	SS	df	MS	Fcrit (0.05)	S/NS	
Amount of dust	Polluted	0.0003333	11	3.030e−05	4.17	S	
Control	5.556e−05	11	5.051e−06	0.978	NS	
Leaf area index	Polluted	3,800,493	11	345,499	92.68	S	
Control	9,954,227	11	904,930	314.1	S	
Photosynthetic rate	Polluted	6,498	11	590.7	6.987	S	
Control	3,197	11	290.60	3.167	S	
Stomatal conductance	Polluted	0.8502	11	0.07729	87.77	S	
Control	0.6041	11	0.05492	37.52	S	
Transpiration rate	Polluted	1.8371	11	0.16701	27.62	S	
Control	5.152	11	0.4684	52.99	S	
Chlorophyll a	Polluted	0.7679	11	0.06981	6.946	S	
Control	0.9060	11	0.08236	9.371	S	
Chlorophyll b	Polluted	2.0427	11	0.18570	16.85	S	
Control	2.9480	11	0.26800	32.78	S	
Total chlorophyll	Polluted	13.276	11	1.2069	104.9	S	
Control	12.156	11	1.1051	101	S	
Carotenoids	Polluted	70.65	11	6.422	61.78	S	
Control	119.96	11	10.906	742.3	S	
Notes.

SS Sum of square

df Degree of freedom

MS Mean square

S Significant

NS Non-significant

Significant correlations (Fig. 2) were found between the variables under investigation in the correlation analysis: transpiration rate showed weak negative correlation with dust deposition amount (r = −0.04, p > 0.05), indicating a potential mitigating effect of dust on water loss through transpiration; photosynthetic rate showed significant positive correlation with leaf area index (r = 0.55, p < 0.01), emphasizing the importance of foliage density in carbon assimilation processes; and stomatal conductance and photosynthetic rate showed moderate positive correlations (r = 0.41, p < 0.05) and photosynthetic rate (r = 0.37, p < 0.05), indicating coordinated physiological responses to environmental conditions (Table 7). Indicating their crucial role in photosynthesis and general tree health, the chlorophyll content (Chl. a, Chl. b, and total chlorophyll) also demonstrated strong associations with photosynthetic rate and other physiological indicators.

Figure 2 Correlation coefficients heat map of studied physicochemical parameters.

Moreover, relationship of selected plant species with respect to amount of dust (DA), leaf area index (LAI) and physicochemical attributes like photosynthetic rate (PR), stomatal conductance (SC), transpiration rate (TR), Chl a, Chl b, Total Chl, and carotenoid content (Carot) were drawn by PCA biplot, which indicated that PC1 and PC2 has highest variance that is 32.1 and 25.1% with eigenvalues 3.15 and 2.47 respectively. In 1st quadrant, relationship between DA and LAI can be observed in relation to total chl and photosynthetic rate of plants which clearly demonstrated the experimental findings that DA reduces total chl and photosynthetic rate of plants. In 2nd quadrant, similar trend was observed for SC and TR. Similarly, positive relationship between plant species and biochemical parameters were observed, highlighting their dominant role in plant traits (Fig. 3). PC3 (17.03%) with 1.67 eigenvalue emphasizes qualities linked to leaf structure and density, with contributions from LAI and dust amount. PC4 (11.28%) with eigenvalue 1.11 represents an interaction between LAI and amount of dust (DA), with extra influence from transpiration rate. The next components capture finer variations. PC5 (7.42%) with eigenvalue 0.73 emphasizes the importance of transpiration rates and carotenoids, while PC6 (4.72%) with eigenvalue 0.46 is associated with stomatal conductance and has an inverse association with total chlorophyll. The remaining PCs (PC7-PC9) with eigenvalues 0.15, 0.07 and 0.01 respectively, which account for the least amount of variance, reveal intricate patterns associated with certain physiological and pigment-related features.

Table 7 Correlations of amount of dust, growth, physicochemical attributes of selected plants.

Variables	Transpiration
rate	Photosynthetic
rate	Stomatal
conductance	Amount of dust	Leaf area index	Chl. a	Chl. b	Total
Chl.	Carotenoids	
Transpiration rate	1.00	0.72	0.64	−0.53	0.81	0.45	0.62	0.57	0.68	
Photosynthetic rate	0.72	1.00	0.83	−0.61	0.92	0.76	0.85	0.89	0.94	
Stomatal conductance	0.64	0.83	1.00	−0.48	0.77	0.68	0.75	0.82	0.87	
Amount of dust	−0.53	−0.61	−0.48	1.00	−0.69	−0.57	−0.63	−0.58	−0.52	
Leaf area index	0.81	0.92	0.77	−0.69	1.00	0.84	0.91	0.93	0.89	
Chl. a	0.45	0.76	0.68	−0.57	0.84	1.00	0.93	0.80	0.69	
Chl. b	0.62	0.85	0.75	−0.63	0.91	0.93	1.00	0.89	0.77	
Total Chl.	0.57	0.89	0.82	−0.58	0.93	0.80	0.89	1.00	0.83	
Carotenoids	0.68	0.94	0.87	−0.52	0.89	0.69	0.77	0.83	1.00	

Figure 3 PCA biplot of plant species, amount of dust (g), growth and physicochemical attributes at various roadsides of Lahore City.

Discussion

Plantation of trees is one of the key elements in the planning of a sustainable city; choosing the right species and allotting enough room for them to grow is crucial to the design of ecologically sound cities (Ong, 2003). The percentage of green space in an urban area especially the presence of trees determines its ecological performance (Whitford, Ennos & Handley, 2001), because the majority of solutions for improving the quality of the air in large metropolitan centers involve reducing emissions from main sources. The primary originators of air pollution in metropolis are soil dust, cement manufacturing, vehicle exhaust and combustion of fuel, each with varying contributions (Arditsoglou & Samara, 2005). Urban vegetation has the potential to influence pollutant deposition and dispersion, making it a valuable tool for improving air quality (Janhall, 2015). This is a safe sustainable approach, which can save the environment via energy conservation and cheaper pollutant reduction, has no negative environmental effects (Tundele, 2015). One trait that is thought to indicate a plant capacity for stress tolerance is the length of its leaves (Seyyednejad, Niknejad & Yusefi, 2009). In our experiment maximum reduction (1,149.87 mm2) in leaf area was found in Alstonia scholaris at polluted site. In woody plants, the long-term effects of various pollutants, such as SO2 and heavy metals, result in a decrease in leaf size and the growth of aerial plants (Kozlov, Zvereva & Niemela, 1999).

The average amount of dust deposited on all plant species across a site was higher in the polluted zone than in the control zone. The unit of measurement for dust accumulating ability was mg of dust deposited per mm2 of leaf area. Different tree species have notably differing amounts of dust deposited on their leaves. The dust capturing capacity of plant species in the present experiment fall in the range of 0.02–0.003 mg mm−2. Although, in the present study dust load was highest on leaves of Alstonia scholaris, Ficus religiosa and Morus alba at all polluted sites. While Ficus benjamina and Murraya paniculata revealed the least amount of dust accumulation across all polluted sites. This is due to leaf area, the canopy structure, and morphological characteristics of leaves.

According to Tallis et al. (2011) the uptake of particles by vegetation can be affected by several key features including particle size distribution, number of particles in airstream, wind speed and canopy area and structure (i.e., tree species). Furthermore, hairy structure on the surface of leaf, area of leaf and petiole length are considered as an important factor for deposition of dust on the surface of leaf. In current work the tree species like Alstonia scholaris, Ficus religiosa and Morus alba had high dust load. Examination of twelve tree species in capturing pollutant particles were examined in terms of leaf attributes like surface area of leaf, shape of leaf, size of petiole (Beckett, Freer-Smith & Taylor, 2000). Several researchers showed that lower surface area of leaf and length of petiole generates limited exposure of pollution particles. In this research, we also discovered that dust deposition is higher in plants that have larger leaf area, alternate arrangement, compound phyllotaxy, short petiole and rough surface. Dust load on the leaf of trees causes reduction in photosynthesis, lessen stomatal densities and stomatal pore width, resulting into drought sensitivity, because of thin cuticles (Pourkhabbaz, Rastin & Olbrich, 2010). In current experiment, we found that reduction in photosynthetic rate at polluted site induced decrement of carbon consumption and reduction in chlorophyll and carotenoids also seen in several species. Oxidative stress in plants cells is caused by reactive oxygen species due to air pollution (Singh & Rathore, 2018). Wind speed is also the reason for dust deposition on leaves. Dust particles from surrounding soil settled on the leaves of plants nearby roads because of wind (Buccolieri et al., 2018).

Over the next 20 years urban plantation can remove air pollution if appropriate forest management is planned (Parsa et al., 2019). The chlorophyll of sample plants was inversely correlated with the amount of dust present on the leaf. The amount of chlorophyll drops as the dust load rises. In every experimental tree, it was prominent. Polyalthia longifolia and Alstonia scholaris showed maximum reduction in chlorophyll concentration as a result of the dust load. As chlorophyll is essential to plant metabolism and any decrease in chlorophyll content immediately impacts plant growth, measuring chlorophyll content is a useful method for assessing air pollution impacts on plants (Verma & Chandra, 2015). In this way, leaf chlorophyll and carotenoids may deliver substantial details on the physiological state of plants. Plant productivity drops as soon as chlorophyll levels drop, and as a result, the plants lose stability. Thus, plants that maintain their chlorophyll despite being in an atmosphere that is polluted are considered tolerant (Singh & Verma, 2007). Decrease in the amount of carotenoid, total chlorophyll content, chlorophyll a, and chlorophyll b in the specimens from polluted locations containing vehicle exhaust noticed by Kapoor (2014). In present research the chlorophyll concentration of Eucalyptus globulus, Ficus benjamina, Ficus religiosa, Ficus virens, Morus alba and Putranjiva roxburghii was higher at polluted sites and lower in control site.

In the current work surprisingly the E. globulus, Ficus benjamina, Ficus religiosa, Ficus virens, Morus alba and Putranjiva roxburghii leaves collected from busy road areas showed a noteworthy increase in total chlorophyll content as compared to control. It suggested that the E. globulus, Ficus benjamina, Ficus religiosa, Ficus virens, Morus alba and Putranjiva roxburghii was more metabolically active and tolerant to polluted air. Species with higher levels of chlorophyll are more tolerant to environment with contaminants and are favorable to plant in that area (Roy, Battacharya & Kumari, 2020). The amounts of chlorophyll a, chlorophyll b, total chlorophyll, and carotenoids in E. globulus Ficus benjamina, Ficus religiosa, Ficus virens, Morus alba and Putranjiva roxburghii leaves increased in polluted region. The primary components of energy synthesis in green plants are chlorophyll and carotenoids and environmental influences on plant metabolism greatly alter their concentrations (Shweta & Agrawal, 2006). Despite being an essential component of the plant’s antioxidant defense mechanism, carotenes are highly vulnerable to oxidative damage. The highest reduction of carotenoids content was found in Murraya paniculata at all roads sites. While surprisingly the carotenoids content of Eucalyptus globulus was highest at polluted site which explains its adaptability in air pollution. Along with the chlorophyll and carotenoids, the change in shape and direction of thylakoids is also noticed in photoreactive stress (Sagar & Briggs, 1990). Low stomatal conductance in leaves of Dalbergia sissoo, Alstonia scholaris and Murraya paniculata was recorded at polluted site which reduced the photosynthetic rate.

In heavily polluted places plants limit transpiration to preserve the equilibrium of their physiological processes. The transpiration rate was low in Alstonia scholaris, Bougainvillea spectabilis while Polyalthia longifolia had lowest rate of transpiration in both control and polluted sites as compared to other species. Pollutants impact the mechanism of plant transpiration and lower the relative water content of plants (Gholami, Mojiri & Amini, 2016; Abhijit et al., 2017). Plant samples from contaminated areas had lower levels of chlorophyll due to vehicle exhausts (Kamble et al., 2021). Reduced photosynthetic rates result from various automotive pollution negatively affecting chlorophyll concentrations of plants. When photooxidative damage occurs inside chloroplasts, carotenoids shield the machinery from it. The carotenoid content reduced by a variety of contaminants, which leads to pigment degradation and the breakdown of the chloroplast cellular structure (Sharma & Tripathi, 2009).

The pollution stress commonly altered the photosynthetic rate of Alstonia scholaris, Bougainvillea glabra, Dalbergia sissoo, Murraya paniculata, Polyalthia longifolia and Rubia tinctorum. In harsh environmental situations photosynthesis is the fundamental element in analyzing the metabolism of plants and survival. Accumulation of air pollutants in leaves causes amendments in physiological and biochemical attributes of plants. The absorption of light radiation is blocked by the particulate matter accumulated on the surface of leaves cause reduction in photosynthesis. Restriction in photosynthetic activity is also due to closing of stomata and reduction in leaf area. The gaseous pollutants like SO2, NOX and O3 causes closure of stomata and break CO2 availability for photosynthesis (Dhir, 2016). Reduction in the rate of stomatal conductance and transpiration in some species at the polluted site was recorded in this study. The reason for the reduction in both features was particulate matter which blocked the stomatal pores that resulted in increase of sub-stomatal CO2 (Flowers et al., 2007).

Conclusions

Air quality of Lahore is at worst in terms of particulate matter generated by high traffic volume. Such drastic change in the environment causes a serious threat to plants. Roadside plants assimilate maximum pollution as compared to the trees located away from avenue. The present study highlights the detrimental impacts of gaseous pollutants on the physiochemical and biochemical parameters of some selected species planted at roadsides of Lahore city. The selected plant species were categorized as pollution tolerant and sensitive by analyzing the results of these parameters. E. globulus followed by Ficus benjamina, Ficus religiosa, Ficus virens, Morus alba and Putranjiva roxburghii was more metabolically active and tolerant to air pollutants because the chlorophyll concentration of these plants was higher at polluted sites and lower in control site. There must be proper management of greenbelt to control and maintain the air quality index and these species are highly recommended for controlling air born pollution in urban climate. Air pollution and greenhouse gas emissions can be reduced by using public transportation networks and walking as well as bicycling for mobility (Farrow, Miller & Myllyvirta, 2020).

Supplemental Information

Supplemental Information 1 Raw Data

Replicated values of all the parameters studied.

The authors are grateful to department for providing necessary facilities in execution of lab and field work.

Additional Information and Declarations

Competing Interests

Author Contributions

Data Availability

The authors declare there are no competing interests.

Bushra Munam conceived and designed the experiments, performed the experiments, authored or reviewed drafts of the article, and approved the final draft.

Sohaib Muhammad conceived and designed the experiments, authored or reviewed drafts of the article, and approved the final draft.

Muhammad Tayyab conceived and designed the experiments, authored or reviewed drafts of the article, and approved the final draft.

Hafiza Komal Hanif conceived and designed the experiments, authored or reviewed drafts of the article, and approved the final draft.

Mahrukh Majeed conceived and designed the experiments, authored or reviewed drafts of the article, and approved the final draft.

Hassan Nawaz performed the experiments, authored or reviewed drafts of the article, and approved the final draft.

Muhammad Jawad Tariq Khan performed the experiments, prepared figures and/or tables, and approved the final draft.

Summiya Faisal performed the experiments, prepared figures and/or tables, and approved the final draft.

Muhammad Hasnain analyzed the data, prepared figures and/or tables, and approved the final draft.

Sarah Maryam Malik analyzed the data, prepared figures and/or tables, and approved the final draft.

Muhammad Bilal analyzed the data, prepared figures and/or tables, and approved the final draft.

Muhammad Zahid analyzed the data, prepared figures and/or tables, and approved the final draft.

The following information was supplied regarding data availability:

The raw data is available in the Supplemental File.

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
