# Peer review of "Physiochemical screening of road avenue plants in better landscape management of highly polluted urbanized city (Lahore), Pakistan"

_PeerJ, doi:10.7717/peerj.20121_

## Round 0.1 · original submission · Major Revisions

· Academic Editor

Major Revisions

Based on the comments from four anonymous reviewers, your article requires major revisions before it can be accepted. Note the PDFs from 3 of the reviewers.

·

Basic reporting

Physiochemical screening of road avenue plants in better landscape management of highly polluted urbanized city (Lahore), Pakistan

1. The name of the article is suitable;
2. The Abstract is well written;
3. Line 99 - Uk, Belford & Hogarh, 2019 – it is Uka;
4. Line 597 - Wei et al – the year is 2021 (it’s written 20221);
5. The conclusions are well-constructed and informative;
6. Under Table 2 is given a Legend - * and ** - it is not clear what exactly means – the range of probability? – Please provide more precise information!

Experimental design

Well done, methods of investigation are with references.

Validity of the findings

Good.

Additional comments

No

Reviewer 3 ·

Basic reporting

Overall, this manuscript presents a comprehensive study on the 'Physiochemical screening of road avenue plants in better landscape management of highly polluted urbanized city (Lahore), Pakistan'.The research structure is reasonable, the methods have certain scientific validity. This research has a certain application value. However, there are some issues that need to be addressed. It is worth publishing after revising these issues:

L21 Abstract:The abstract should reflect the research method.

L47 Introduction: Part of the logic is a little confusing. It is suggested that the global part be mentioned before the Lahore city part.

L54 “In the last decade, Lahore has been regarded as one of the most polluted cities in the world.” Reference should be added, and specific information on the current state of pollution should also be identified.

L111 The article mentions that “For various plant species, ascorbic acid represents oxidative stress and is required for cell wall development and cell division.”,but ascorbate peroxidase (APX) was not measured in the study. If the measurement and analysis of APX index were added, the research results should have stronger persuasiveness and support.

L140 The article mentions that the basis for selecting 7 main roads as research sites is traffic volume. Please add to the specific traffic data.

L165 Please clarify the specific time of “the day's peak rush hour”

Tables 2-5 and Figures 1-3 are not clearly pointed to in the text. Please add the links in the text and provide details about the tables and figures.

Experimental design

no comment

Validity of the findings

no comment

Reviewer 4 ·

Basic reporting

Thanks a lot to the Authors for working on the time demanding issues of the air pollution and probable solution with the appropriate vegetation. However, they did not maintain the professional scientific structure to make this manuscript worthy for publication. More specifically, they could not relate the necessary justification of this study in relevance with the stated objectives.

Experimental design

The experimental design is not enough to full fill the objectives with necessary methods stated in this manuscript. Moreover, there is lacking of the statistical approaches to make the results in acceptance with substantiate the required statistical analysis. However, the authors visualized some information regarding the correlation matrix, PCA-biplot etc. but they did not explain anything about these in the results section even in the discussion section as well.

Validity of the findings

The findings of this manuscript are not successfully validated with the current figures and tables. Moreover, the authors did not follow any statistical tool to statistically justify their findings, only used the mean value that is not sufficient enough to make any sense in a scientific manner.

Additional comments

I have mentioned the specific comments on the respective areas of the attached reviewed manuscript for better clarification to the authors. The conclusion section and abstract section almost similar but that should not be. Authors failed to write the conclusion in a scientific manner focusing on their objectives. Moreover, their selected objectives were seemed very general not specific and achievable as well.

Annotated reviews are not available for download in order to protect the identity of reviewers who chose to remain anonymous.

·

Basic reporting

- The abstract is too qualitative, whereas in the report there are some quantitative data. It would be better to take some most important quantitative data to support the qualitative interpretation of the data.
- The structure of Introduction should be revised.
- The explanation about the street name for sampling is not giving proper information regarding the position of each streets. It would be better if provided the map of the city
- It would be better if in the result, the performance of each plants is reported under categorized of parameters measured, in that way it would be easier for reader to compare parameters of each plant.
- Data in tables and pictures should be elaborated in the result
- The result should be compared with other related reports and discussed properly.

Experimental design

- The explanation about the street name for sampling is not giving proper information regarding the position of each streets. It would be better if provided the map of the city
- Line 151: spring season is the peak season for plant growth? Not at summer season? Please give a reference
- Line 151: commonness after? What does it means?
- Line 161-162: explain the “entirely random design”
- Line 169: leaf area, length, width and LAI is not physiological parameters
- The method section also need to be rewritten thoroughly

Validity of the findings

- It would be better if in the result, the performance of each plants is reported under categorized of parameters measured, in that way it would be easier for reader to compare parameters of each plant.
- Data in tables and pictures should be elaborated in the result
- Line 203-205 repetition from method
- Line 211-212 how to keep an eye on atmospheric pollution?
-

Additional comments

- The result should be compared with other related reports and discussed properly.

---

## Round 0.2 · accepted · Accept

· Academic Editor

Accept

I confirm that the authors have addressed all the reviewers' comments and the revised manuscript is ready for publication.

Reviewer 4 ·

Basic reporting

No comment

Experimental design

No comment

Validity of the findings

No comment

Additional comments

Dear Authors, thank you so much for your tremendous efforts to revise the manuscript and make necessary improvement for the worthy of publication in the prestigious journal like Peer J. Please check the discussion section of the attached file for minor corrections and the figure2 and table 7, these are duplicate of data presentation you should keep one. Check comments on the attached reviewed pdf file.

Annotated reviews are not available for download in order to protect the identity of reviewers who chose to remain anonymous.

·

Basic reporting

The manusciprt has been revied properly following my previous comments, and I am satisfied with the revisions made.

Experimental design

The experimental design has followed the sciientific standard in this area

Validity of the findings

The finding of the experiment is meaningful and encourage coparison in other area.

Additional comments

I agree that the manuscript is ready to be published in this journal